# DNS Tunnelling, Exfiltration and Detection over Cloud Environments

**DOI:** 10.3390/s23052760

**Published:** 2023-03-02

**Authors:** Lehel Salat, Mastaneh Davis, Nabeel Khan

**Affiliations:** 1Faculty of Engineering, Computing and the Environment, Kingston University, Penrhyn Rd., Kingston upon Thames KT1 2EE, UK; 2Department of Computer Science, University of Chester, Chester CH1 4BJ, UK

**Keywords:** DNS tunnelling, DNS exfiltration, the elastic stack, DNS monitoring, cloud computing, AWS, GCP, Iodine, DNScat2

## Abstract

The domain name system (DNS) protocol is fundamental to the operation of the internet, however, in recent years various methodologies have been developed that enable DNS attacks on organisations. In the last few years, the increased use of cloud services by organisations has created further security challenges as cyber criminals use numerous methodologies to exploit cloud services, configurations and the DNS protocol. In this paper, two different DNS tunnelling methods, Iodine and DNScat, have been conducted in the cloud environment (Google and AWS) and positive results of exfiltration have been achieved under different firewall configurations. Detection of malicious use of DNS protocol can be a challenge for organisations with limited cybersecurity support and expertise. In this study, various DNS tunnelling detection techniques were utilised in a cloud environment to create an effective monitoring system with a reliable detection rate, low implementation cost, and ease of use for organisations with limited detection capabilities. The Elastic stack (an open-source framework) was used to configure a DNS monitoring system and to analyse the collected DNS logs. Furthermore, payload and traffic analysis techniques were implemented to identify different tunnelling methods. This cloud-based monitoring system offers various detection techniques that can be used for monitoring DNS activities of any network especially accessible to small organisations. Moreover, the Elastic stack is open-source and it has no limitation with regards to the data that can be uploaded daily.

## 1. Introduction

The Domain Name System (DNS) protocol is fundamental to the operation of the Internet with the primary purpose of translating domain names to IP addresses. This important service facilitates access to the Internet by using website names or any arbitrary domain names. The DNS protocol communicates internal requests to external remote servers and creates a landscape for attackers to perform malicious activities such as stealing credentials and sensitive data [1,2,3]. One of the most effective DNS attack types is DNS tunnelling, where the attacker creates a tunnel between the client’s network and the attacker’s machine, utilising command and control channels to obscure data and bypass the firewall and intrusion detection systems [1,4]. Recent research by PaloAlto [5] indicates that 80% of malware attacks gain access to the command and control channel by leveraging the DNS protocol. EfficientIP [6] claimed that in 2020, 79% of organisations experienced a DNS attack, either directly or a DNS communication was part of the attack.

The use of cloud services has become more embedded within businesses, as cloud platforms provide a cost-effective, flexible and scalable environment. This is evidenced by the continuous growth of cloud services for software or infrastructure in the last 10 years [7,8,9]. Due to the challenges of the COVID-19 pandemic, organisations continued by working off-premises and services such as e-mail systems, cloud servers, VPNs and advanced cloud services facilitated this digital transformation for businesses [10]. However, the flexibility of cloud services introduces increased complexity for customers, who should not assume that the security settings are optimised for their particular environment. It is important to note that the security of a cloud platform follows a shared responsibility model, where cloud service providers and customers must monitor and respond to threats.

Monitoring DNS queries for unusual data and DNS domains can be an effective method [11], but this process is commonly overlooked by information technology (IT) and security administrators [12]. DNS is easy to identify and is a well-defined protocol, typically operating on standard ports. However, the volume of DNS traffic coupled with unusual data types can provide a challenge [13]. Some cloud providers use public DNS over HTTPS (DoH) to preserve user’s private data. The use of this security method is complex and can make DNS data monitoring more challenging for organisations and users [14,15].

There are numerous methodologies and approaches for DNS data collection, monitoring and detecting DNS tunnelling. The majority of detection approaches are based on statistical analysis of network traffic and packet content (payload analysis) [12,16], or machine learning/deep learning techniques [4,17,18]. These detection strategies (specifically machine and deep learning techniques) may produce false positive results, such as blocking legitimate domains, as highlighted in [2].

This is a challenging area for businesses and users due to their limited access to state-of-art systems that are typically proprietary platforms mostly affordable to larger organisations. The present study proposes a cost effective and accessible solution using an integrated open-source monitoring system in the cloud environments.

### Contributions of the Present Study

In this work, a cloud-based approach has been adopted to generate DNS tunnelling traffic and statistical analysis of DNS data in cloud environments, where the victim’s machine and attacker’s server have been implemented in Google and AWS cloud platforms, respectively. In summary, the contribution of this work is as follows:The majority of the research work on DNS tunnelling is conducted over traditional architecture. One of the main contributions of this work is the demonstration and detection of DNS tunnelling and exfiltration in cloud environments.Simulate DNS tunnelling and detect attack indicators in the cloud environment. Demonstration of using the Google Cloud platform for setting firewall rules and blocking all outbound (egress) traffic, the DNS still resolves IP addresses via metadata server and, as a result, a DNS tunnel can be successfully set.To show the importance of monitoring DNS data and to establish a monitoring server in a cloud environment for real-time detection of DNS tunnelling and exfiltration. Furthermore, several approaches have been analysed in the monitoring server for detecting exfiltration of data via DNS. We have demonstrated the most efficient strategies (in terms of accuracy of detection, ease of implementation and the use of open-source modules) for the detection of DNS tunnelling and exfiltration.

This paper is organised as follows. Section 2 presents the background and information about DNS tunnelling approaches and detection techniques. Section 3 reports the adopted methodology for the demonstration of DNS tunnelling and exfiltration techniques in the cloud environment. Section 4 and Section 5 describe the experiments in the cloud-based environment and a DNS tunnelling connection between an attacker and a victim under different firewall constraints using port 53. Section 6 provides details of the monitoring server implementation and, in Section 7, different detection techniques are explored and discussed. Finally, Section 8 provides a detailed discussion of the findings and proposes a direction for future research.

## 2. Related Work on DNS Tunnelling Detection

A DNS tunnel can be established by using a client-server connection, where resolving DNS servers on the client side will act as proxies for the tunnel. There are a number of tools [18] which can be used for setting a DNS tunnel, such as, DNScat2 (https://github.com/iagox86/dnscat2 (accessed on 20 March 2021)), Iodine (https://code.kryo.se/iodine/ (accessed on 20 March 2021)), etc. The author in [13] designed an experiment where the client and the malicious server were set as virtual machines and implemented a DNS tunnel using the Iodine and DNScat2 tools. In this study, live DNS data was investigated by analysing the payload and traffic data by using the statistics engine of the Splunk framework. The proposed setup detects malicious domains by deploying statistical analysis of the payload and traffic data such as length and entropy of the fully qualified domain name (FQDN), uncommon resource record types and the volume of DNS queries.

The payload assessment can be achieved by analysing several DNS protocol attributes such as DNS query content, domain name length, specific signatures and the hostname entropy. Farnham [12] also presented DNS traffic analysis from the properties such as the geographic location of the DNS server, frequency and volume of DNS requests, and number of hostnames per domain. According to the analysis, a domain with a large number of sub-domains as unique hostnames will be considered as an anomaly.

Another feature of the DNS protocol is the resource record type, which presents the type of correspondence, e.g., type A for IPV4, type AAAA for IPV6, NS (name server) record of authoritative server for a domain, TXT for text query, null for arbitrary content, etc. The common record types (such as A, AAAA, MX, CNAME, NS and TXT) are more frequently transmitted DNS queries, hence, statistical analysis of the records and the frequency of less common record types such as null can provide an assessment strategy to identify DNS tunnelling [13,19,20].

The authors in [20] presented a DNS monitoring strategy for unencrypted traffic in which common records such as A and AAAA are statistically analysed. The authors claim that recent DNS tunnelling attacks are mainly developed based on A and AAAA records. They generated DNS data by using tools such as Cobalt strike to create a sample data with A and AAAA records. After DNS data creation, the authors investigated anomaly detection strategies based on measuring the amount of information in the DNS traffic and randomness (entropy) of the traffic data. Their analysis showed that DNS tunnelling attacks can be accurately detected from A and AAAA records rather than analysing subdomains and TXT records. However, their proposed approach is unable to identify DNS tunnelling over encrypted traffic such as DNS over TLS and HTTPS (DOH) protocols. Although the majority of DNS works through the transmission of plain text, the usefulness of the DOH protocol, which can protect the privacy of DNS queries and responses, may provide an opportunity for malicious activities to hide through encryption [21].

The work in [22] developed a detection system where the victim’s server, attacker server and Elasticsearch have been set in a virtualised environment. Iodine and DNScat2 tools were used to generate DNS tunnel data and collected DNS traffic using packetbeat, which was then analysed by Elasticsearch and visualised by Kibana. They detected DNS tunnelling using the number of hostnames per domain, where a domain with large number of unique hostnames will be considered as an anomaly.

Another important monitoring technique is based on passive DNS data [2]. The passive data is based on known threats and historical data, which can be used to uncover potential security incidents or discover malicious domains and networks. This approach led to several studies [3,23,24] for detecting various DNS attacks. These studies analysed large passive DNS datasets and identified DNS features which have been used for detecting malicious DNS domains and usage. The usage of passive DNS in identifying DNS attacks is widely researched by many authors [19,25], however, solely relying on existing datasets/sources may not be deemed suitable as attackers continue to use novel attack methodologies. Furthermore, collection of passive DNS data requires a lot of resources such as open source information gathering (domain, subdomain, IP address etc.), processing and analysing collected information.

Table 1 presents a summary of relevant studies on DNS tunnelling monitoring methods in non-cloud environments including their weaknesses and advantages. The data set column provides information related to data that was used or built for each study.

## 3. Methodology

To perform DNS tunnelling, two cloud providers were selected as shown in Figure 1. The DNS tunnel server (attacker) was based on the Amazon Web Service (AWS) cloud and a Webserver (victim) was hosted in the Google Cloud platform (GCP). A monitoring server based on the Elastic stack was also hosted in the GCP. It is important to note that a separate virtual private cloud (VPC) in GCP hosts the monitoring server and the client’s webserver, where both servers were implemented on two different subnets as shown in Figure 1. Each VPC has a firewall, and firewall rules are applied to all the VM instances hosted within the VPC. The VPC network runs a local metadata server https://cloud.google.com/vpc/docs/firewalls#gcp-metadata-server (accessed on 5 December 2022) alongside each VM instance. The metadata server provides basic services such as DHCP and DNS name resolution. The metadata server resolves both internal and external DNS queries, regardless of any firewall rules, using Google’s public DNS servers.

The experiments were conducted over the period of eight days, i.e., the webserver was accessible from the internet for eight days. The majority of the DNS traffic was generated from the attacker server and a small proportion of traffic was produced from non-malicious domains such as security updates from debian.org and browsing google.com. All information related to the DNS traffic was sent to the monitoring server. The size of the data ingested by the monitoring server was approximately 3.6 megabytes of DNS queries and 4 megabytes for responses. The implementation details of the monitoring server are provided in Section 6, whereas setup of the DNS tunnel server (attacker) is provided below.

To perform exfiltration of data from the webserver via DNS in the cloud environment, two different tunnelling applications, Iodine and DNScat2, were tested. Iodine tunnels IPv4 data through the DNS tunnel server. It is specifically used in situations where Internet access is firewalled but queries via DNS are allowed. On the other hand, DNScat2 creates an encrypted command and control (C&C) channel over the DNS protocol. To use both these applications, two actual domains, slehee.com and slehee.uk, were purchased for the purpose of tunnelling and exfiltration of data from the webserver. It is not in the scope of this paper to detail how an attacker can gain access to the victim machine. Therefore, we assume that an attacker has managed to compromise the target system (either via social engineering or through exploitation of a vulnerability) and deployed the tunnelling software on the victims machine (webserver hosted in GCP).

### DNS Resolution Paths

DNS is a hierarchical system, and exfiltration can be executed if a system DNS server is configured to allow an external request to upstream DNS servers. For this experiment, as the victim had never visited the malicious domains, slehee.uk and slehee.com, the IP addresses of them needed to be determined. Figure 2 demonstrates several steps for this process which can be summarised as below:

The tunnelling application on the client-side sends out a DNS query to the local DNS server to determine the IP address of the domain as shown by Step 1 in Figure 2.Since it is not in the local DNS server records, the request will be forwarded to one of the root DNS servers (Step 2).The root servers will forward the request to one of the top-level domain (TLD) servers that are responsible for the .com and the country domains. In this work these are the slehee.com and slehee.uk domains, respectively (Step 3).The local DNS server queries the TLD server as shown by Step 4 in Figure 2.Finally, TLD refers to the authoritative nameserver of slehee.com (Steps 5, 6 and 7), which has been modified to point to the AWS instance IP address. Hence, the client query (IP address of slehee.com) is resolved by the local DNS server (Step 8) as shown in Figure 2.

The following sections demonstrate two experiments with different approaches for tunnel creation between the webserver and attacker instances hosted in two different cloud providers.

## 4. Experiment 1: No Firewall Constraints on Port 53

In the first experiment, the domain slehee.com was used to establish a direct connection from the client machine (webserver in GCP cloud) to the malicious server in the AWS cloud. The client machine will either establish a connection directly to the malicious server or query the authoritative DNS server, which will point to the nameserver of the attacker machine. To start the DNS tunnelling and exfiltration attack, the tunnelling server on the AWS instance initiates the iodine service as shown in Figure 3. To verify if the system is set up correctly, Iodine provides a test page. Figure 3 shows the test result and that the Iodine server side is up and running and is accessible through ss.slehee.com (Step 1 in Figure 3).

For this study, GCP as an infrastructure as a service (IaaS) has been used where the cloud provider is responsible for providing services such as servers, storage, and networking resources. Therefore, by default, the GCP firewall allows users to use the DNS services, and hence port 53 is always open.

Once the webserver was infected by the Iodine application, a DNS query was initiated by the webserver as shown in Figure 3 (Step 2).

The query was received by the DNS server in GCP, which consulted the DNS architecture of the Internet (root and TLD servers as shown in Figure 2) to resolve the IP address of the attacker’s domain (Step 3 in Figure 3). The GCP DNS service resolved the query of the webserver by returning the IP address of the attacker’s server (Step 4 in Figure 3). After the DNS query was resolved, the attacker’s server was able to establish an SSH connection through the DNS tunnel and start the exfiltration process. According to Figure 3 (Step 5), the content of the /etc/passwd file, along with other files, were sent from the client machine (webserver in the GCP instance) to the AWS instance. Highlighted in red, it can be observed that the passwd file from the client was successfully sent over to the server at 14:26 pm.

## 5. Experiment 2: Tunnelling Connection under the Constraint of Firewall Blocking Port 53

In experiment 2, a DNS tunnelling attack was conducted under the constraint of the cloud firewall blocking all the traffic from the webserver to the Internet (egress traffic). The implementation of this experiment was similar to Experiment 1 apart from the firewall constraint. Figure 4 shows the public and the private IP addresses of the AWS instance (attacker’s server). The two nameservers, ns1.slehee.uk and ns2.slehee.uk, pointed to the tunnelling server instance in AWS. The Iodine service started at 3.141.76.210 on the AWS instance (slehee.uk) as shown in Figure 5. It is important to note that URL to DNS resolving is active at the webserver in the GCP, even though the firewall configuration is blocking all egress traffic from the webserver, as shown in Figure 5. The process of tunnelling is initiated once the webserver gets infected with the Iodine payload. All the steps of tunnel initiation (DNS look up by the DNS, root and TLD servers) are the same as in Figure 3. Figure 5 highlights the process of successful data exfiltration via DNS tunnelling under the limitation of the firewall blocking all the outbound traffic of the client machine hosted in the GCP.

In contrast to this experiment, if the webserver is hosted locally, not in a cloud environment, then the firewall restriction of blocking all the egress traffic would have been sufficient to block the DNS tunnelling attack. To demonstrate this key concept and highlight the difference between the two environments (cloud vs locally hosted), a webserver with a pfSense virtual firewall was locally deployed and all the egress traffic was blocked. The pfSense is also configured as a DNS resolver. The configured firewall blocks DNS requests from local clients to servers outside the local network. In other words, clients are forced to send DNS requests to the DNS resolver on pfSense. This is achieved by creating two rules https://docs.netgate.com/pfsense/en/latest/recipes/dns-block-external.html (accessed on 15 January 2023). The first rule allows DNS queries to the pfSense firewall (pass rule: pass DNS to the firewall), whereas the second rule blocks DNS (Deny rule: block DNS to everything else). Therefore, access to other DNS servers on port 53 will not be possible. This is shown in Figure 6, where the Iodine connection attempt for DNS tunnel initiation was unsuccessful mainly because port 53 was blocked by the firewall. However, as shown in Figure 5, the DNS tunnelling is successful in the cloud environment as Google cloud runs a local metadata server alongside the webserver instance, which provides basic services such as DNS, DHCP and network time protocol. This server is fundamental to the webserver’s functionality; hence it is not possible to block the DNS resolution path through the firewall configuration. This can be addressed by implementing an alternative DNS name server (or host-based firewall) combine with the creation of an exclusion and inclusion list of domains. However, until the domain or the corresponding IP address is blacklisted [26], the DNS connection will be successful in the cloud environment.

In the aforementioned experiments, Iodine was used as the tunnelling application. DNScat2 is another tunnelling application, which creates an encrypted command and control channel between the attacker and the victim. Figure 7 shows a successfully encrypted channel (via slehee.uk domain) created by utilising DNScat2. According to the figure, the server (attacker) and the client machine (victim) present the same secret message which shows that the connection was successfully established.

After the demonstration of DNS tunnel initiation and exfiltration of data in the cloud environment as given above, the next section discusses an approach through implementation of a monitoring server for the detection of malicious DNS activities.

## 6. Monitoring Server Implementation

The monitoring server was based on the Elastic stack (https://www.elastic.co/what-is/elk-stack) (accessed on 23 January 2021) (formerly known as the ELK stack) and deployed in the GCP. The stack is a collection of open-source tools for managing complex data. The main parts of the stack are Elasticsearch, Logstash, Kibana and beats.

The monitoring server was set on a different subnet (subnet 2) than the webserver, which was set on subnet 1 as reported in Figure 1. The internal traffic on the subnets was not restricted between the instances and were established by the default router. The default firewall rules were extended for the ELK Kibana application for monitoring purposes from the admin side only. In this experiment, the DNS traffic was captured by installing Filebeat and Packetbeat for the DNS tunnel client machine (webserver). The DNS traffic was stored and analysed on the monitoring server. Suricata is an open-source intrusion detection system with several sets of public rules that are community maintained and applied to the data imported from the client machine. Suricata has a separate module for the Filebeat package, which was enabled, and the log location added to the Filebeat module in the webserver client. The DNS log files are in the .txt format, however, the Suricata module parses DNS logs in the JSON format, i.e., the DNS logs are sent and received in the JSON format. It is important to note that Suricata rules are applied to all the DNS queries (malicious and non-malicious) imported from the webserver. The malicious DNS queries and responses, which were detected by Suricata and the Elastic stack, were analysed and discussed in Section 7.

### 6.1. Overhead of the Elastic Stack Based Monitoring

The financial overhead of the proposed setup, in terms of processing burden and implementation cost, is minimal. It is important to note that the beats framework of the Elastic stack provides light-weight data shipping capabilities. The Packetbeat tool ingests network data to the Elasticsearch, whereas the Filebeat extracts and ships data from applications and system logs of the webserver to the Elasticsearch module. Both beats modules ingest data in real-time, and require minimal computing resources in the webserver. The monitoring server is equipped with Elasticsearch, Kibana and Suricata. There is neither licensing cost nor vendor related restrictions associated with the use of these modules. The Elastic stack along with Suricata provides an accessible, real-time, cloud-based, and cost-effective solution for adoption by individual users, and businesses [27].

### 6.2. Elastic Stack Security

To take full advantage of the opportunities of the stack and further secure it, TLS/SSL was applied on the whole stack to encrypt the communication between the log shippers, Elasticsearch and Kibana. To enable encryption across the whole stack the following steps were applied:A private key and X.509 certificates generation for each node.Nodes in the stack were configured to use the signed certificates for authentication.The monitoring server was configured to use an encrypted connection.Kibana on the monitoring server was configured to encrypt communication between the server and the browser and to establish a connection to Elasticsearch via HTTPS.All beats were configured to use an encrypted connection.

## 7. DNS Tunnelling Detection

This section presents the evaluation of the DNS data that was collected from the experiments described above. The two DNS tunnel servers (Iodine and DNScat2), which were implemented in the AWS cloud environment and the webserver (set in the Google cloud environment) browsed the DNS tunnel server, and establish some basic activities such as updating the server to generate DNS traffic. The DNS traffic was collected at the webserver, and transported to the monitoring server in the JSON format.

To identify the DNS tunnelling activities, the generated DNS logs were stored and grouped into payload and traffic data categories. The collected data was examined by utilising different detection techniques such as statistical analysis of the DNS payload, and analysis of uncommon record types. In addition, several other techniques such as assessment of the DNS traffic data from the volume of DNS queries, the volume of DNS traffic per domain, and number of hostnames per domain were deployed. The details of these strategies are discussed in the following subsections.

### 7.1. Statistical Analysis of Domains and Subdomains

Most legitimate domain names tend to have a meaningful human-readable name, while domain generated algorithm (DGA), malware and tunnelling applications use randomly created domains and subdomains [28,29]. Hence, domains with a higher randomness may indicate a DNS attack. The randomness of information can be measured by the concept of information entropy or Shannon entropy, which was introduced by Claude Shannon [30]. The Shannon entropy, H, of a discrete set of probabilities $p_{i}$, is defined as
(1)$$H=-\sum_{i=1}^{n}p_{i}\times \log\left(p_{i}\right)$$
where *n* is the total number of observed events, i.e., i=1,2,3,…,n. The Shannon entropy definition and score can be used for domain and subdomain names of DNS queries where higher entropy values will indicate a high randomness of a domain.

In this study, the DNS logs gathered by the Elasticsearch and the randomness of ss.slehee.com and ns.slehee.uk domains have been identified by extending the search interval to include the DNS tunnelling activities from Iodine and DNScat2. One of the features in Elasticsearch is statistical analysis of the strings, which can be set to determine the Shannon entropy of the queries. Shannon entropy measures uncertainty of the given domain name and it provides fast and accurate results in real-time. If the entropy value exceeds a defined threshold score, then this may be an indication of DNS tunnelling activities. The results of this experiment show a high entropy score for ns.slehee.uk, and a high-level of randomness in the queried domain names, as shown in Figure 8. Generally, entropy scores of non-malicious DNS queries are less than 4 [12]. According to Figure 8, the entropy score of more than 4 for all the considered DNS queries highlights the exfiltration of data via the DNS protocol. Typically, normal and safe domains have a lower entropy score; for example, the entropy value for the google.com and debian.org domains were obtained and measured approximately 2.5.

However, this approach may result in false-negative outcomes, as there are some exceptions, where DNS names are used to represent information, e.g., version names where numbers and characters are included in the queries or domain name generators that are often used by large cloud providers. Another benign element which can cause higher than normal entropy is content delivery network (CDN) assistance and site hosting [31]. In such scenarios, the character frequency probability may help to identify the nature of a DNS query. For this study, in order to analyse the probability of each character, the DNS data packets were monitored using the packetbeat platform, which is integrated into the Elastic stack.

Analysis of the *dns.question.subdomain* field reveals that some subdomains have a lot of numerical characters in the DNS queries. To get a more accurate view of the character distribution, the Elastic developer tools were used to analyse the *dns.question.subdomain* field with the Shannon entropy value based on the frequency probability of each character as shown in Figure 9. This figure also shows the distribution of the characters probability in descending order, where numerical characters are residing at top of the list. According to the figure, a higher entropy value coupled with a higher integer frequency probability is a strong indicator of data exfiltration by exploiting the DNS protocol.

Another important feature is the length of the DNS queries, which in turn shows the number of characters (bytes) in the subdomains and can be used to detect DNS tunnelling and exfiltration activities. It is important to note that the encoding method (base64 or base16) greatly impacts the amount of exfiltrated data. Base16 encoding can exfiltrate half bytes of data (per ASCII character), whereas base64 exfiltrates 0.75 bytes of data per ASCII character in the subdomains. Both encoding methods utilised for data exfiltration result in longer query length. Therefore, the length of the subdomains employed in the exfiltration of data is an important indicator. Figure 10 shows the query length for the ns.slehee.uk subdomain, which was setup for the DNS tunnelling experiment, and the average subdomain length for all queries. The query length was 122 characters with an average subdomain length of 46, indicating a high standard deviation which is an indicator of an anomalous DNS query. In another words, the length of malicious query (used for data exfiltration) was more than double the length of average queries over the last seven days of the experiment.

### 7.2. DNS Record Type Analysis

The next DNS tunnelling detection strategy is based on analysing DNS record types. This monitoring technique can be effective as there are only a few DNS record types that are used for DNS tunnelling. Figure 11 illustrates record types in the DNS queries. Iodine utilises the null record for data exfiltration via a DNS tunnel. Each DNS reply can contain over a kilobyte of compressed payload data. The significant percentage of null requests shown in the bar chart represents the Iodine DNS tunnelling activities. Another important point to consider is that the null record was deprecated by RFC1035. Therefore, a significant proportion of null queries is a strong indication of malicious DNS exploitation.

### 7.3. DNS Signature Analysis

Another popular detection method is based on analysing the DNS signature, which can be used to review specific attributes in the DNS header and DNS payload content [12]. The authors in [16] demonstrated a detailed signature analysis in which the DNS response packet from null records was exported and saved as a binary file by using Wireshark, as shown in Figure 12. The binary file was then analysed by using the Neo Hex Editor and detection was implemented by using Snort rules, for instance the Snort rule to identify malicious network activity is SID-1-27046 (https://www.snort.org/rule_docs/1-27046) (accessed on 20 March 2021), which is based on the Iodine DNS tunnelling handshake server ACK. Analysis of the collected null record using Wireshark is shown in Figure 12.

Another important feature of DNS data is time-to-live (TTL), which describes how long a DNS response for a domain should be cached and can be used for detection of DNS tunnelling. The recursive resolvers usually cache the information so instead of retrieving them again from the authoritative server, can simply reply with the cached record. The duration that the DNS record is required to be kept in the cache of a DNS server is referred to as TTL, which for a normal DNS record is mostly greater than 300 s. Hence, DNS data with a TTL value close to zero may indicate DNS tunnelling activity as a malicious user can use this approach to avoid DNS detection and blacklisting. Figure 12 shows that the DNS signature has content matches all highlighted in red, where the hex value 0x0001 indicates a null record and the TTL value of zero indicates no time for the intermediate cache, which points to the existence of DNS tunnelling [24,32].

The Suricata open-source intrusion detection and prevention system (IPDS) can also be used in the cloud environment for creating rules to identify a DNS tunnelling attack. Figure 13 shows the Suricata alerts during DNS exfiltration with Iodine in ELK-stack. The Suricata IDPS can be combined with other alert rules such as suspicious null request. Similarly, other alerts can be created to detect increased TXT records. Moreover, the detection success can be increased by taking advantage of the Elastic stack SIEM, where new rules can be created alongside the 400 default rules.

### 7.4. DNS Traffic Volume and Time Analysis

An effective strategy for traffic analysis is based on the fact that limited data can be exfiltrated in a DNS tunnel. Therefore, an increase in volume of DNS traffic over time may indicate malicious DNS activity. This strategy will be effective for networks with expectations of a lower number of DNS queries. Figure 14 shows high DNS traffic that was captured from the cloud environment experiments. However, in networks with a high volume of DNS traffic, this method will most likely give false positive results. Therefore, in the case of networks where the number of DNS queries is expected to be low, a high volume of DNS traffic may indicate DNS tunnelling activity.

Another detection strategy is based on a given domain (suspicious domain) with multiple subdomains [12], where a suspicious domain with multiple subdomains can be a strong indicator of DNS tunnelling. In this study the ss or sl and ns subdomains are short, and a large amount of data was ingested in each of the subdomains. During the exfiltration, the client–server communication was hidden in the domain or subdomain names and, as result, a large number of new unique subdomains were generated. Figure 15 shows a significant number of subdomains for slehee.com and slehee.uk. In addition, the right-hand side table in Figure 15 shows a large number of bytes that are generated from DNS queries and responses (bytes in and out). This demonstrates the presence of malicious activities.

Another interesting observation for exfiltrated traffic via DNS is a higher time interval between consecutive pairs of queries and responses. This can be seen from Figure 16 where the average response time during exfiltration is approximately 70 ms (shown using green colour). The average response time for normal DNS traffic is approximately 20 ms [32]. Therefore, a higher than normal average response time is a strong indication of DNS tunnelling activities.

## 8. Discussion

This study proposed a cloud-based detection solution, which was tested by analysing the collected real time DNS data. Two experiments in the cloud environment were conducted to demonstrate the importance of monitoring DNS traffic to identify DNS tunnelling attacks.

The results of the monitoring strategies tested highlight that without suitable measures to monitor network traffic, DNS tunnelling applications can quickly establish connections to the command and control server, bypassing the outbound firewall restrictions. For the proposed cloud-based detection system, the Elastic stack has been configured successfully and, in order to evaluate the performance, the following methodologies were utilised: Shannon entropy of hostnames, character frequency probability, hostname length, number of subdomains, increased DNS request, unusual records type, TTL and DNS signature. For analysing DNS traffic data, the ELK-stack beats modules such as packetbeat and filebeat were implemented to provide meaningful data logs.

The signature analysis results prove that Suricata could be a valuable detection tool when applying detailed and specific rules. The experiment to determine the randomness in the domain names and subdomains with Shannon entropy proved successful. The possibility of obtaining false-positive indicators will increase as more applications move towards the cloud environment. Therefore, the query for domain-generated algorithm detection needs to be fine-tuned to filter out trusted domains. For instance, it has been shown that the use of Shannon entropy along with character frequency probability analysis can filter out trusted domains. Furthermore, it has also been shown that the calculation of average response time will improve the DNS data monitoring system as this will enhance the detection accuracy even for DNS over HTTPS flows.

The demonstrated detection techniques show the effectiveness of the open-source framework of the Elastic stack. Furthermore, the implementation of such a framework requires minimal effort, which makes it possible for enterprises with limited resources to fight against the most common cyber threats. Whilst these approaches have been used before in non-cloud environments, this work successfully demonstrates that malicious queries can be accurately identified in a cloud environment by utilising these techniques. Together, these methods can provide a useful detection strategy which can be contained in one dashboard.

For future development, the DNS tunnel detection strategy can be enhanced by adding passive DNS monitoring. Passive DNS monitoring has the potential to identify malicious domains using multiple IP addresses. This can aid real-time detection, thus uncovering the IP footprint of the DNS tunnelling attack. Furthermore, there are some Elastic stack SIEM prebuilt rules that can be implemented to improve the detection of malicious activities as part of the regular monitoring of the DNS traffic. For example, rules can be configured to detect DNS activities when an internal network client sends DNS traffic directly to the Internet, or when it receives abnormally large DNS responses.

## Figures and Tables

**Figure 1 sensors-23-02760-f001:**
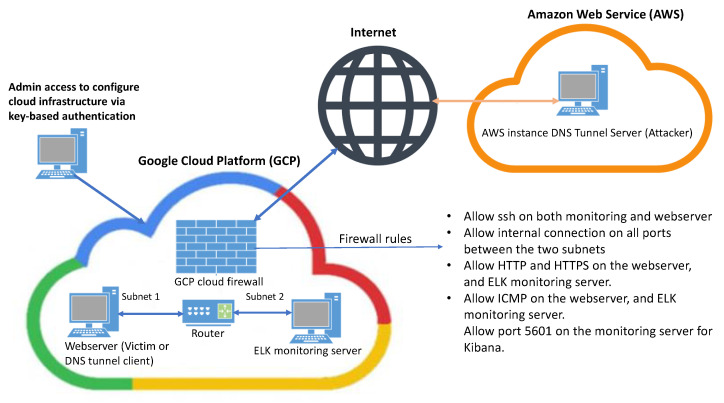
Implementation details of DNS tunnelling and exfiltration of data setup in the cloud environments.

**Figure 2 sensors-23-02760-f002:**
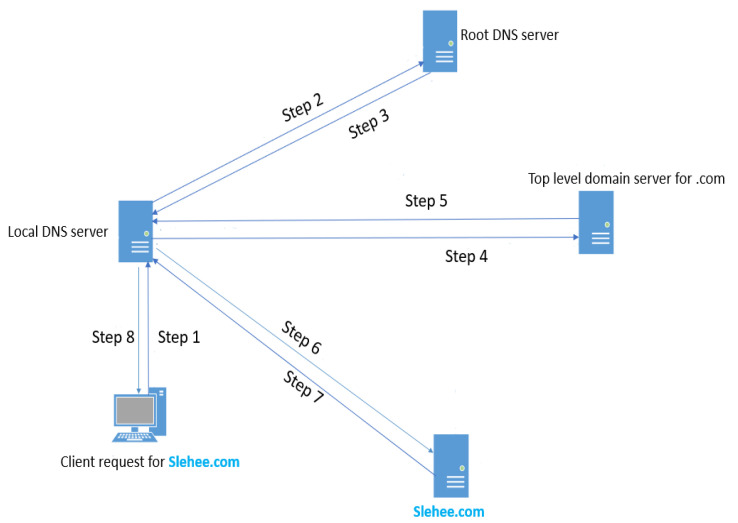
Resolution path of the DNS hierarchical system.

**Figure 3 sensors-23-02760-f003:**
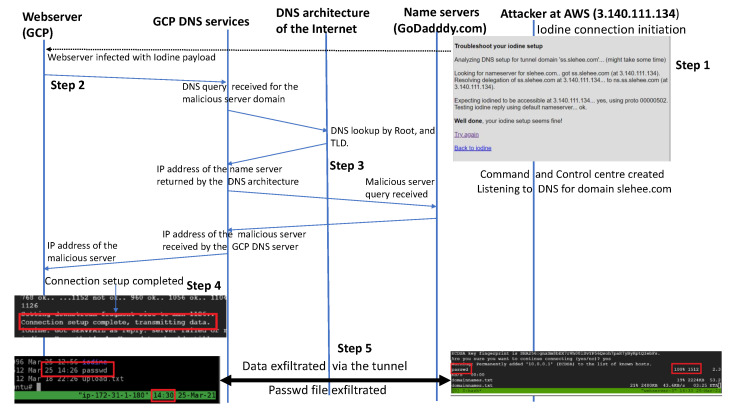
Sequence diagram showing DNS tunnel initiation and data exfiltration.

**Figure 4 sensors-23-02760-f004:**
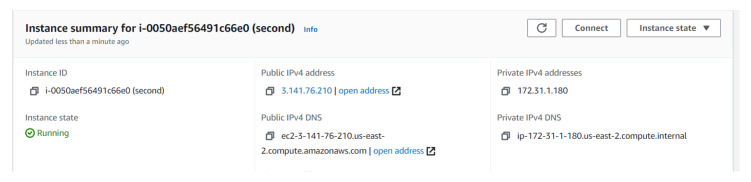
Experiment 2: AWS instance public and private IP addresses.

**Figure 5 sensors-23-02760-f005:**
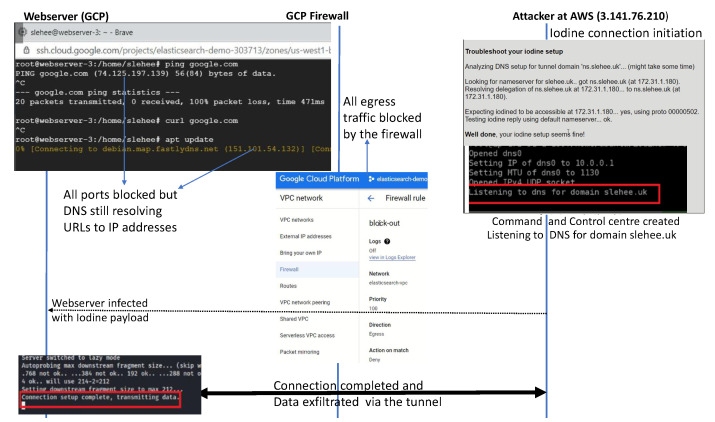
Experiment 2: Sequence diagram showing DNS tunnel initiation and data exfiltration under the constraints of firewall blocking all the ports.

**Figure 6 sensors-23-02760-f006:**
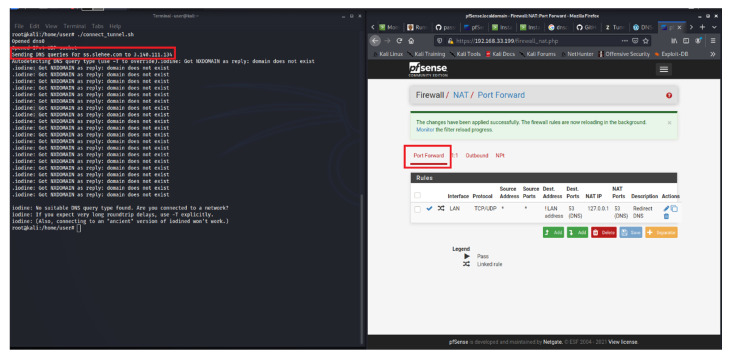
Webserver hosted in a non-cloud environment: Iodine connection failed because port 53 is blocked by the virtual firewall.

**Figure 7 sensors-23-02760-f007:**
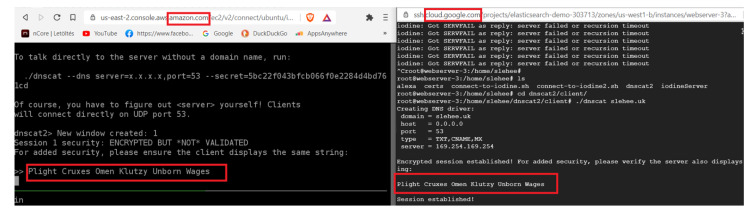
DNS tunnelling connection established by utilising DNScat2.

**Figure 8 sensors-23-02760-f008:**
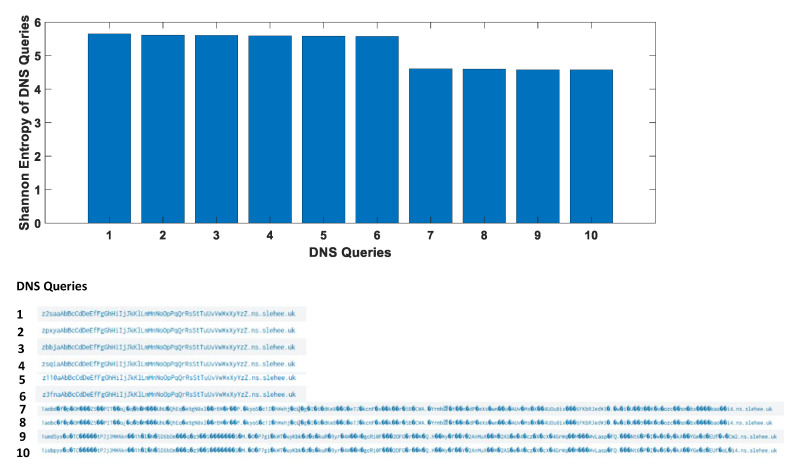
The first part of the figure shows Shannon entropy score of DNS queries. The second part of the figure lists the DNS queries.

**Figure 9 sensors-23-02760-f009:**
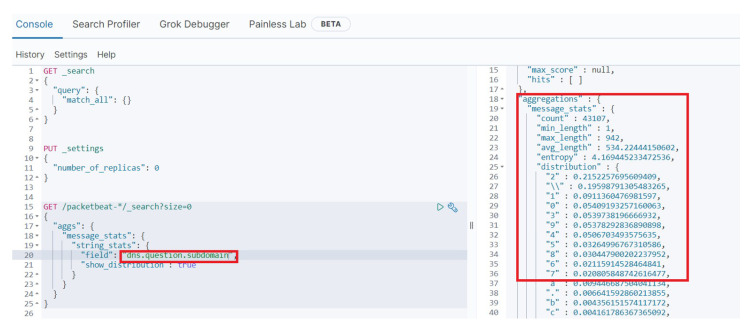
Character frequency probability visibility shown in the Kibana visualisation dashboard.

**Figure 10 sensors-23-02760-f010:**
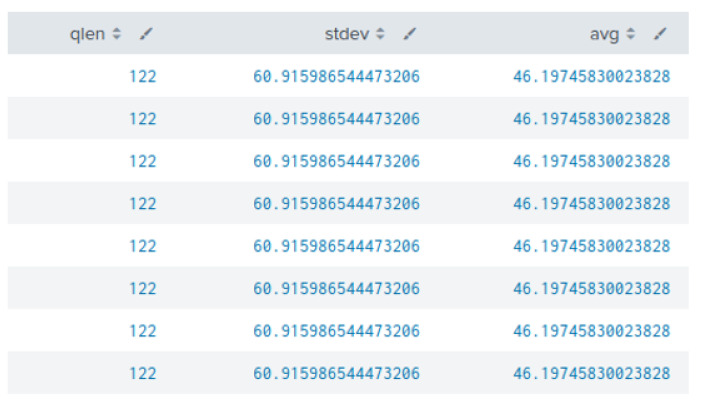
First column shows the DNS query length for the ns.slehee.uk subdomain. The second column shows the standard deviation between the elements of the first and second columns. The third column lists the query length averaged over the last seven days.

**Figure 11 sensors-23-02760-f011:**
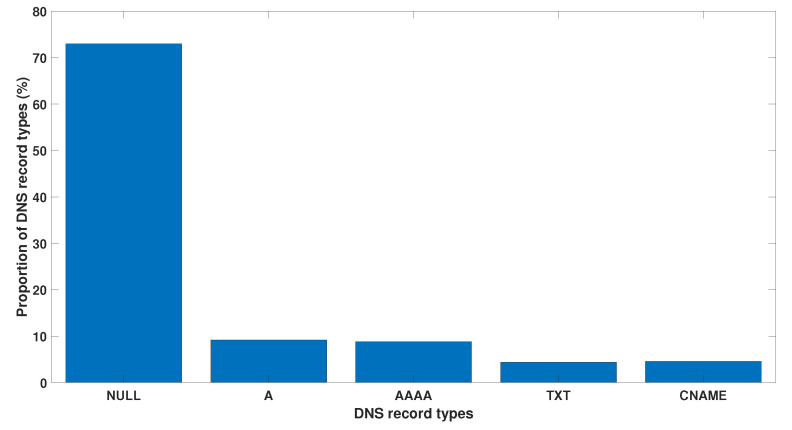
Proportion (%) of the different record types in the DNS queries.

**Figure 12 sensors-23-02760-f012:**
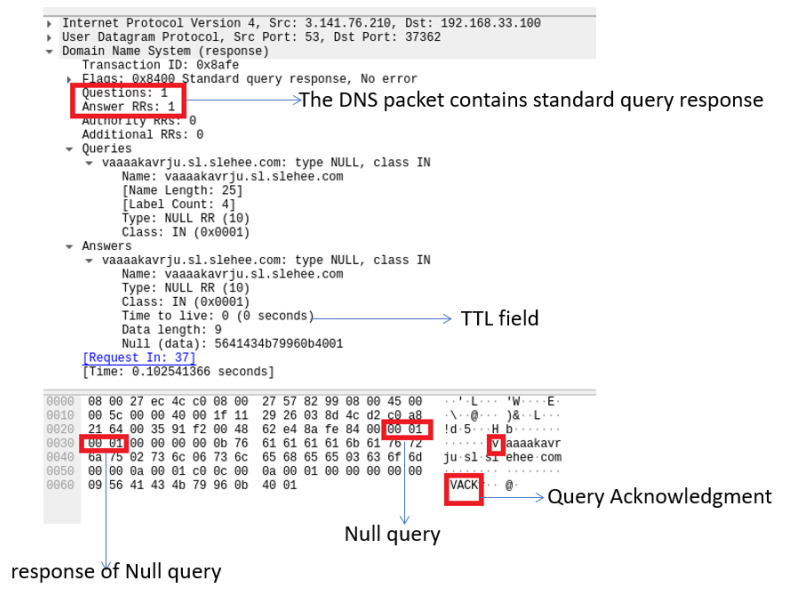
Handshake (ACK) between Iodine DNS tunnelling server (attacker) and the victim. Analysis via Wireshark shows the DNS response packet from Null record exported and saved as a binary. The figure also shows the TTL field in the DNS query response.

**Figure 13 sensors-23-02760-f013:**
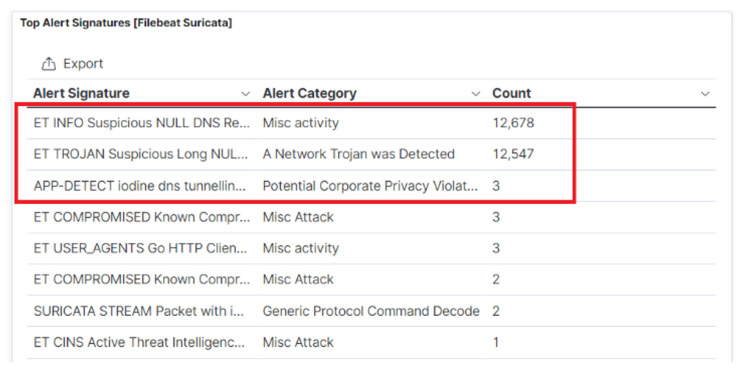
Suricata alerts in the Elastic stack during DNS exfiltration with Iodine.

**Figure 14 sensors-23-02760-f014:**
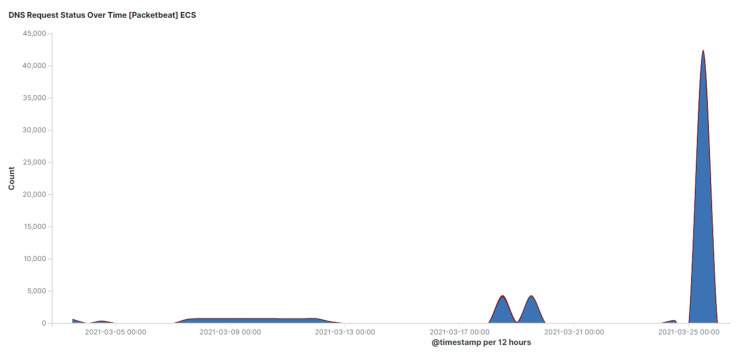
DNS traffic request volume over time.

**Figure 15 sensors-23-02760-f015:**
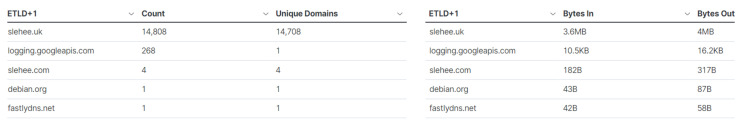
Unique subdomains per domain shown in the Kibana visualisation dashboard.

**Figure 16 sensors-23-02760-f016:**
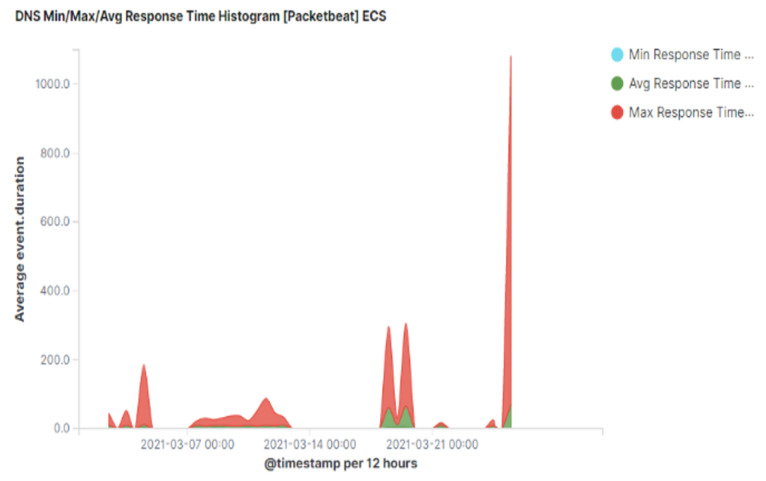
Average time interval between DNS query and response.

**Table 1 sensors-23-02760-t001:** Summary of the relevant studies for DNS tunnelling detection methods in the non-cloud environment.

Paper	Detection Methods	Dataset	Weaknesses	Advantages
[19,24]	Statistical analysis of resource records type and usage of new fully qualified domain names on a passive DNS data.	Passive DNS	Collecting passive DNS data requires resources. Identifying DNS tunnelling through filtering passive DNS data can be unreliable.	Accurate results when combined with other detection strategies (such as statistical analysis).
[16]	Statistical analysis of DNS traffic using Snort IDS.	Malicious data generated through Iodine.	Relatively challenging to write a good rule for detection and can produce false positive alerts. Packet processing can be slow.	Snort IDS is an open-source tool; flexible, scalable and easy to set.
[13]	Statistical analysis of DNS payload and traffic, analysing the length and entropy of DNS requests using Splunk.	Malicious data generated through Iodine and DNScat2.	Splunk is a costly platform.	Built in state-of-the-art data-analytics tools.
[20]	Statistical analysis of A and AAAA records only.	Malicious data generated through Dns2tcp, DNScat, DNScapy, Cobalt strike, pisloader, Dnsdelivery and Glimpse.	The strategy does not consider other record types, such as txt, CNAME, and null	Low processing overhead because of considering only two record types.
[22]	Analysis of DNS traffic, using Elasticsearch in a virtualise environment.	Malicious data generated through Iodine and DNScat2.	The Elastic stack implementation can be complex as it requires setup of multiple open source tools. The work only considers number of hostnames per domain for detecting DNS exfiltration.	Use of an open-source platform, i.e., Elasticsearch for DSN tunnelling detection, Kibana dashboard for visualisation and beats framework for data export.
[17,18]	Machine and deep learning-based strategies	Malicious data generated through Iodine, DNScat and Ozyman.	Implementation of this method is complex and availability of bigger datasets for training can be a problem.	Automated process with high detection accuracy.

## Data Availability

Not applicable.

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
