# Peer review of "DNS Tunnelling, Exfiltration and Detection over Cloud Environments"

_sensors, 2023, doi:10.3390/s23052760_

Round 1

Reviewer 1 Report

-The authors introduced two different DNS tunnelling methods in the cloud environment and provided the positive results of exfiltration under different configurations of firewall. They illustrated DNS tunnelling detection techniques having high detection accuracy, low implementation cost in cloud environments, and ease of use for organizations with limited cyber support.

-The following corrections are required:

Section 1: Introduction

-The introduction section is too general, and it introduces concepts that are well known about the DNS tunnelling and exfiltration in cloud environment. The introduction does not stimulate to go ahead with the remaining of the paper because it does not introduce any really new topic/solution. Furthermore, "the research motivation" of paper at the introduction section is missing.

-Besides, the authors describe some of works about the DNS tunnelling and exfiltration in cloud environment, while some of the papers that should have be included are:

https://www.sciencedirect.com/science/article/abs/pii/S0306457322000528

https://link.springer.com/article/10.1007/s00500-020-05409-2

https://ieeexplore.ieee.org/abstract/document/9886602

https://onlinelibrary.wiley.com/doi/abs/10.1002/spe.2641

https://ieeexplore.ieee.org/abstract/document/9644728

https://link.springer.com/article/10.1007/s10586-019-02972-8

--In addition, a conclusion of related work in the forms of a table in terms of evaluation tools, utilized techniques, performance metrics, datasets, advantages, and disadvantages could reconcile from other researchers work to the own one.

Section 2: Methodology

-Please provide a sequence diagram to show the interaction between components of the proposed DNS tunnelling in the cloud environments according to Figures 2-4.

-What is the overhead proposed solution? Please provide a subsection to discuss about the overhead of proposed.

Sections 3-7: Experiments

-The evaluation is incomplete. I would like to see an evaluation on the proposed solution in terms of execution time under different scenarios.

-Paper needs some revision in English. The overall paper should be carefully revised with focus on the language: especially grammar and punctuation.

- Figures of the experiments are in low quality and hard to read when printed.

-Overall, there are still some major parts that the authors did not explain clearly. Some additional evaluations are expected to be in the manuscript as well. As a result, I am going to suggest Major revision the paper in its present form.

Reviewer 2 Report

DNS tunnelling, exfiltration and detection over cloud environments

Comments:

This paper replicates prior work demonstrating the effectiveness of two tools for creating DNS tunnels that could be used for the exfiltration of data from a compromised host. After successful demonstration, there is an analysis to identify such traffic.  While the validation of prior work is sound, the novel approach the paper offers in its title isn’t delivered. There is no substantive explanation as to why the experiment should differ in cloud environments.  The subsequent analysis again applies a number of established techniques.  The use of ELK is interesting as a visualisation dash, but this could have been explored more. 

Some items to consider in the research are, consideration of the potential configuration of the selected tools, to try be 'less obvious'. An example of this is the NULL record which to most operators would not be something seen on a production network - especially since it was deprecated by RFC1035 in 1987! (isn’t legacy wonderful).  As other work indicates A/AAAA records are the most plentiful and are a reasonable place to hide. TTL is another consideration which seems to have been omitted. Given the constraints of FQND paths, its desirable that these do not stick around in intermediate caches longer than necessary, and as such tunnelling often is indicated by the factors raised in combination with lower TTL values.   The entropy detection within domain names is interesting, and there is a large body of work looking at the DGA's within malware and the identification of these within traffic streams (DNS included)

While the packetbeat collector running on the client is reasonable in the experimental setup (and if one has an isolated cloud server), a better placement would be at a gateway, or in the case of an internal /controlled resolver on the client side of that.  Anything that can be done to reduce the execution of privileged code where it is not needed should be done.

Where the paper fails to deliver is anything fundamentally novel to the field of network security (or even sensing) an existing setup is replicated on cloud hosts. The paper also has the feeling of having been cut down form a larger draft, and this may contribute to some of the sections not flowing well.  This is exacerbated by a a significant number of typographical errors and inconsistencies between diagrams and text. The reference sin particular are not at a level that would be expected for submission for publication. Diagram quality could be significantly improved, and the author(s) should aim for a more unified look across these. Where possible, quantitative data should be presented in such a way is is meaningful and useful to other researchers.

If the authors were able to address some of the inconsistencies above (and detailed below) the paper would be in much better shape. I would suggest trying to rebalance the focus of the paper. much is done about the setup of the initial experiments, but the actual analysis /detection is rather rushed.  The relevance and significance of the 'cloud' needs to have a stronger motivation

Page 1 line 31 - Be clear that this is a threat in general, not to the DNS protocol or operation.

35 - space between 80 %

45 - software updates at scale should be related to known sources.

51 Why is [2] cited as A Khormali, there is only one in the paper, others do not use initial eg [9]

54 - Inconsistency in [7], [8], and Bilge et al [9] why is the later 'named’? Consider Bilge being a new sentence

57 - authors, [10], [11], extra commas

59 - Justify the claims around expense, and privacy. BE clear what is meant by passive data. Consider mentioning existing datasets/sources which are deemed not suitable.

69 - Could the discussion around false positives be expanded to quantify the kinds of cases (or refer reader to suitable citation)

71-74 - The paragraph switches context midway though.  The sentence relating to tunnels needs to be moved (or at least split the paragraph here), and the narrative around the data sources and collection closed out.

75 [5] has a single author.

85 Null vs NULL earlier - consistency is important.

87 where *resolving* DNS servers

101 - Reads awkwardly  wrt little information, consider revising.

103 - Reads better as traffic rather than traffics

134 - host based firewalling is another viable solution for restricting traffic and DNS in particular enforcing that only set server(s) are used.

157 - Sections

173 - inconsistent naming /capitalisation of dnscat2 though the document.

Page 5 - consider the sizing of Fig 1 &2

Fig 1 - there is a red 'typo' artefact in the bullet point mentioning ssh

Fig 3 - consider redrawing the client ->internal dns connection to be clear that the request is relayed via that system. Consider spacing and sizing.

202 - authoritative server, or its own upstream resolvers.

212 - should this bee sleeh.uk  the remainder refers to the .com

228 - the attacker rather than the 'inanimate' webserver

Figure 5 - the screenshots all refer to the uk domain, mismatch with text ?

Fig 9 Image quality and size

260 - merge the stub sentence: AWS instances. (Figure 4 and Figure 9).

271 - DNScast   -> DNScat (as line 274)

327 subscript i ?

Fig 13 - Is a pie char really the best format. in this case the domain names are not necessarily important and that rather the apparently fairly uniform entropy scored. This could be better implemented as a bar /histogram or even point plot ?

Fig 16 - image quality, again are pi plots really the most meaningful way of communicating the data. would it not be better to label the IPs as you have not discussed all of these. A 'friendly name' may be more meaningful to the reader.

Fig 17 - bearing mind the reader is not likely familiar with DNS packet structure. BE very clear and consider text labels rather than just read bounding boxes , especially in the lower hex decode.

Fig 19 - an interesting area of further exploitation could be the use of homography attacks against popular domains, which could quite likely bypass human operators

eg: legitimate: mdpi.com

attack: ⅿԁрі.ϲоⅿ (URL encoded as: %E2%85%BF%D4%81%D1%80%D1%96.%CF%B2%D0%BE%E2%85%BF)

References:

Line 423- Other papers reviewed for this Journal follow the format of naming the citation list References rather than Bibliography

[2] Khormali et al, why referenced as A Khormali on line 51 ? There is no other Khormali on the reference list.

[3] Inconsistent Capitalisation of title - Using

[6] Title spelled incorreclty

[9] -Missing details relating to where paper published (acm)

[12] errors in capitaliastion around RAID conf

[15,16] inconsistency in citation. latter missing information. Should these mot possibly just be footnotes as they are web pages not actual reports/articles

[19] Which country does the department belong to ?

[20] This should be a footnote rather than a reference, or consider another actual paper for citation

[22] Missing where Shannons seminal work was published.

Reviewer 3 Report

The work is interesting. Presentation and graphics are good and adequate to follow the explanation. Tunneling detection techniques is a topic well established in the bibliography. I think the work can be improved by clearly presenting the improvements over the well established DNS tunneling detection strategies. I think it could be approved with minor corrections.

 The author presents a strategy based on some well established strategies and others indicators as Shannon Entropy for the host name, statistical analysis and use of uncommun record types for detecting DNS tunneling. The author applies this new strategy to a cloud computing environment where attacked hosts and detecting hosts are not in the same subnet. However, packet capture software must be used in the attacked host subnet.

I appreciate the author presenting three experiments in order for the reader to easily understand the type of attack and the strategies used to detect the attacker domain and block it. Also he shows that blocking the DNS port does not stop the attacker. The local DNS redirect the DNS query to the attacker domain.

The work's main contribution is the configuration of new tools to detect DNS tunneling and use of new indicators as Shannon-entropy in a Cloud Environment.

I think the references are appropriate. In the introduction, the author briefly compares his approach with other well stated approaches, but at the end he does not analyze their behavior. We need more information about how this new approach improves other previous approaches. For example, is the use of Shannon-entropy a worthwhile improvement? If yes, how large is the improvement?

There is an error at the end in the "Acknowledgments" section. It appears that it is copied without editing.

Finally, the author briefly discusses the resulting data. How the presented work improves or complements others works about DNS tunneling? The author presents news methods? Are these methods not previously used in the bibliography? Only I found in the text that this is an easy and economic approach because of the use of open software. I think this part could be elaborated a little and the proposed methodology justified in more detail.

Round 2

Reviewer 1 Report

Thanks to authors for the detailed response and additions I read through the comments and skimmed the revised PDF, The updates did improve
the paper a lot. I would be happy to recommend this paper for publication

Reviewer 2 Report

Previously raised issues are largely addressed by the authors.  The paper while improved still has a number of issues that need to be considered.  one of these is being a lot more transparent and explicit around the data collected, and how it was used  (what formats etc) .  There are a still a  number of places where the language needs to be cleaned up. Thin particular there is a prevalent issue of quite long sentences. Another issue is the lack of the definite article “the”.

These aside, the work describes a series of experiments, that while not novel are valid.

I still fail to see the direct relevance to the cloud. The ‘loophole’ being describes is pretty much part of almost every LAN design  ( other than home networks) where a local resolver(s) services queries form the local hosts own stub resolvers. thus blocking traffic from lan hosts, and tunnels still persist – as proxied though the local resolvers.  This is exactly what is being applies in the cloud with the exception of a mysterious ‘metadata server’ which is not completely described or qualified.

For these experiments, the authors should be transparent about what the local resolvers (stub or otherwise) on the systems under test are.

There is still no clarity around how environments differ for cloud.  The detection is all moot with the Use of DOH and DoT encrypted transports (other than if looking at an in-path resolver). There is some allusion to the impact of DoH/DoT but   if these are being used to a local or trusted (cloud) resolver then that resolver still ahs insight into the traffic.  I agree that should hosts be using ‘public’ DoE endpoints then things become problematic.

line 30. Start new sentence before EfficientIP; in 20202,that 79% ; be clear if the attack was using DNS (e.g. ddos/interception etc) or DNS communications were part of the attack, there is a subtle but distinct difference.

line 37 the flexibility of

line 43 - can be or is commonly overlooked?

lines 44/45 - DNS is easy to identify, it’s a well-defined protocol, typically operating on well-defined ports, DNS traffic running on nonstandard ports  can also be determined. It is agreed that the volumes can provide a challenge and especially if unusual data types are used.

line 46- this is not really protecting private data (typically considered PII) but rather preserving the privacy of their queries. Remove however and start a new sentence.

line 47 - evidence for this being difficult and complex ? it has become much easier, and many browsers now default to this.  IT is likely not the default. Provide evidence/citation to where this is not the case at a cloud provider.

line 50 tunnelling - then end sentence and start a new one.

line 63 - The majority

lines 66-68 - if all traffic is blocked, then how is this traffic flowing , other than via use of internal DNS servers. This needs to be explicitly stated. the statement as it stands, has a logical flaw.

line 69- "This work" this is a repetition of the listing phrase on line 63.  consider this carefully for improved readability.

line 78 - using port 53?

line 82 performed -> established.

line 83 - resolving servers on the _client_ side

line 84 - the use of the term attack here is provocative, this may well be used as part of an attack, or just for covert communications. Similarly these can be (and are used) for bypassing restrictive networks and censorship. These tools need to be referenced.

line 88 reference needed for Splunk

line 109 attacks

Table 1 - consider the formatting, removing the lines (e.g. with book tabs style) could improve the look substantially. Consider increasing the width of the first column too.

line 136 define safe environment.

line 145, remove spacing before footnotes these should be place earlier when tools are first mentioned. (line 84)

Figure 2 - this could be reduced in size, and still be viable, if the step numbers are enlarged and the thickness of the lines increased.

lines 159-168 - the diagram (fig 2) has unnumbered steps) is numbered, why not refer to these in text and guide the reader though the diagram. As is done in section 4 relating to fig 3.

line 195 – There needs to be clarity  here as to what is defined by external traffic, in this scenario the 'local' dns resolver is still in place. you need to be clear where the firewall is as well. Figure 1 does not indicate where the resolver is. Is the webserver running its own full stack resolver, or just a stub using an upstream?

figure 5 neds to show why this is possible - i.e. what is in resolv.conf? and where is this located. one cannot just wave away this. While the experiment is valid in demonstrating, the authors need to be upfront as to why this is the case (and is commonly exploited -- and in fact is a key part of why DNS tunnels are so popular and effective) as while the client may be blocked, the local resolver is not.

line 209 - this would not be the case if again the DNS resolver was local  ( as is typical for performance and filtering reasons) Figure 6 blocks all traffic from the lan, so what are the resolvers being used in this case.

line 210 pfSense I believe is the correct formatting. non-cloud webserver? This needs to be clarified, the phrasing is awkward.

line 212-214 - this is changing up what is being claimed in the text relating to blocking traffic. here you are explicitly intercepting all traffic for the entire LAN.

line 217 - "metadata server" this must be properly defined. this is *not* common terminology.

line 223 - the Iodine tool -> Iodine

lines 236-237 - a diagram would be good to make sure the reader is clear here with the elements placed and labelled.

line 271 - be explicit as to where/how these logs were generated and stored.

lines 279-284 - there is a body of work around this for detecting fast flux dns with this as one technique. These of this should be referred to. Consider Cleick, Stalmans, Yadav, Alieyan and others.

line 297 - a common benign element that trips up the entropy test is CDN hosts. It would be good to see some validation for example of the Alexa top 1 Million, Majestic million and other 'popular' dns lists.

Figure 8 - the most notable thing here is the non-printable characters. while technically legitimate especially with IDN's they are unusual enough to be detected. a simple test for 'high ascii' would be sufficient. be very clear that these 8 are examples. this may be better presented in a table.  Serial number is the wrong term to be using especially in the context of DNS where it has very different meanings.  It would be good to see some 'normal' and 'safe' domains here as a comparison to demonstrate/validate the claim by [12].

lines 298-302 - citations, link this to the point above.

Figure 10 - you mention 8 days. It is worth being explicit about the period in which the experiments ran. how much data was collected, ( size, space and volume)

Figure 12- response to Null Query. Hex code and using 0x notation is redundant.

line 340 - footnote to the Snort Rule ID ?

line 353 - be clear what snort and Suricate are being run against. This looks to only be 'bad' traffic, what about when there is a mix of traffic introduced for all methods.

Figure 16 - look at the quality of this, there are JPEG artefacts.

References

[5] last accessed date? <- this applies to others as well.

Round 3

Reviewer 2 Report

Changes are done and largely addressed, thanks to the authors for the clear markup and well laid out response.

I’m happy with the way that the metaserver has been explicitly addressed. This of course infers no requirement that it is used for DNS resolution, many operators may opt to run their own stub resolvers pointed to public DNS, or their own full-stack - especially when wanting control over functionality such as DNSSEC. That said I doe believe that many will just go with the default provisioning.

By changing the cloud host to not use the metadata server for resolution, the point is moot.  If upstream DNS servers are being used, these can be restricted. In your local example, you effectively break the network, unless you are operating a separate local recursive/stub resolver, which is in essence the same as the metadata server, with the exception that its Network configuration is fully known.

Fig2 - avoid crossing lines to local DNS server,

Consider carefully if references [7,8,9] are wholly appropriate or add value.

Acknowledgements, this needs to be updated/amended or the boilerplate text removed.
